# F_2_BMF (M = B and Al) Molecules: A Matrix Infrared Spectra and Theoretical Calculations Investigation

**DOI:** 10.3390/molecules28020554

**Published:** 2023-01-05

**Authors:** Juanjuan Cheng, Liyan Cai, Zhen Pu, Bing Xu, Xuefeng Wang

**Affiliations:** 1Shanghai Key Lab of Chemical Assessment and Sustainability, School of Chemical Science and Engineering, Tongji University, Shanghai 200092, China; 2China Academy of Engineering and Physics, Mianyang 621900, China

**Keywords:** inserted complexes F_2_BMF, BF_3_, infrared spectrum, quantum chemical calculations

## Abstract

Reactions of laser-ablated B and Al atoms with BF_3_ have been explored in the 4 K excess neon through the matrix isolation infrared spectrum, isotopic substitutions and quantum chemical calculations. The inserted complexes F_2_BMF (M = B, Al) were identified by anti-symmetric and symmetric stretching modes of F-B-F, and the F-^11^B-F stretch modes are at 1336.9 and 1202.4 cm^−1^ for F_2_^11^B^11^BF and at 1281.5 and 1180.8 cm^−1^ for F_2_^11^BAlF. The CASSCF analysis, EDA-NOCV calculation and the theory of atoms-in-molecules (AIM) are applied to investigate the bonding characters of F_2_BBF and F_2_BAlF molecules. The bonding difference between boron and aluminum complexes reveals interesting chemistries, and the FB species stabilization by a main group atom was first observed in this article.

## 1. Introduction

B_2_ species, comprising electron-precise B-B bonds, have witnessed swift developments in the past twenty years, in which B_2_X_n_ (X = F, Cl, Br, *n* = 2,4) molecules are important precursors for synthesizing boron-containing compounds, which garnered much attention from scientists [1]. As for B_2_F_4_, Trefonas and Lipscomb showed that B_2_F_4_ has a planar structure in the solid phase by X-ray diffraction [2], while several earlier Raman and infrared spectroscopic studies suggested a staggered structure [3,4,5]. In 1977, Danielson, Patton, and Hedberg confirmed that the gaseous B_2_F_4_ molecule has a D_2h_ symmetry by electron diffraction [6]. Fan and Li also found that the ground state of B_2_F_4_ has an eclipsed conformation (in D_2h_ symmetry) [6]. Later, Danielson, Patton, and Hedberg demonstrated that the experimental difficulties in determining the structure of B_2_F_4_ were probably due to its very low internal rotation barrier around the B-B bond (0.42 kcal⋅mol) [7]. For F_2_B_2_, the linear singlet structure FBBF (*D*_∞h_, ^3^Σ^−^) is a second-order stationary point at MBPT(2), and it could convert into a bent structure (*C*_2h_, ^1^A_g_), which is 5.0 kcal/mol higher in energy than a linear structure at MBPT(2) [8]. The double-bond character for the bent structure is suggested by the NBO analysis (WBI-[BB] = 1.432 Å), involving two highest occupied a_g_ and b_u_ MO’s, which can be visualized as a donor-acceptor complex formed by the delocalization of σ lone pairs into empty p_π_ orbitals lying in the molecular plane. The association of two ground-state BF molecules into the *C*_2h_ bent structure has a very low barrier (<1.0 kcal/mol) via a loose *C*_2h_ symmetric transition state (R_BB_ = 2.434 Å) [8]. Recently, our assignment of the experimentally observed B-F stretching frequencies at 1327 cm^−1^ to the trans-bent isomer FBBF is therefore very tentative due to similarity in the calculated anti-symmetric B-F stretching frequencies of the linear and bent isomers [9].

Although B_2_F_4_ and B_2_F_2_ have already been known well, there seems to be no relevant report on the B_2_F_3_ molecule. As we know, B_2_F_3_ should possesse FB_2_ and FB fragments, having significant differences from B_2_F_4_ and B_2_F_2_ molecules, but its structure and properties have not yet been understood. Boron is the main group element in the periodic table, and bonding between boron and main group elements has also been an attractive subject [10,11,12,13,14]. For example, by laser vaporization of a mixed B/Bi target, the Bi ≡ B and Bi = B multiple bonds in BiB_2_O_2_^−^ and Bi_2_B^−^ are observed and are characterized by photoelectron spectroscopy and ab initio calculations [15]. Several years ago, we reported some boryl complexes F_2_BMF (M = C, Si, Ge, Sn, Pb) [14], and DFT and CCSD(T) calculations demonstrate that triplet F_2_BCF is the most stable isomer with two singly occupied molecular orbitals, while singlet F_2_BMF (M = Si, Ge, Sn and Pb) molecules possess a near right angle B-M-F moiety with lone pair electrons on the M atom. In this paper, the laser-ablated B and Al were demonstrated to react with BF_3_ to produce the fluoroboryl complexes F_2_BBF and F_2_BAlF, which have been identified by boron isotopic substitution and theoretical frequency calculations. The bonding formation for B-B as well as B-Al in fluoroboryl complexes of F_2_BMF was investigated by EDA-NOCV and CASSCF calculations. The active molecular orbital and NBO analysis, and the bonding difference, were analyzed in detail.

## 2. Results and Discussion

The assignment of absorptions was based on the behavior of the products’ absorptions upon stepwise annealing and photolysis behavior and will be discussed below. The typical infrared spectra in the selected regions and the absorption bands are shown in Figure 1, Figure 2 and Figure 3 and Table 1, respectively. In addition, the calculated frequencies based on DFT are listed in the same tables for comparison.

### 2.1. F_2_BBF

In the reaction of laser-ablated ^11^B with ^11^BF_3_ in excess neon, as shown in the Figure 1, the new product absorptions upon co-deposition appeared at 1336.9 and 1202.4 cm^−1^. These two bands decreased slightly after annealing to 8 K and 12 K. These two bands shifted to 1370.6 and 1241.6 cm^−1^ in the reaction of ^10^BF_3_ with the ^10^B target, which showed the similar behavior to that of the counterparts produced with ^11^B + ^11^BF_3_.

The 1336.9 cm^−1^ appeared at the BF_2_ stretching region and shifted to 1370.6 cm^−1^ with ^10^BF_3_ + ^10^B, giving the 1.0252 ^10^B/^11^B isotopic frequency ratio, which are in good agreement with the isotopic frequency ratio of 1.0280 in the B_2_F_4_ molecule [3]. The 1202.4 cm^−1^ shifted to 1241.6 cm^−1^, giving the ^10^B/^11^B isotopic frequency ratio of 1.0326 that was close to the calculated ratio of 1.0350 and fit in the previously reported values of the F-B-F vibration mode [16]. Unfortunately, the BBF stretching mode was covered by precursor bands in our experiments. These bands are appropriate for the F_2_B-BF molecule based on the isotopic shifts and photochemical behavior.

In the reaction of ^11^BF_3_ with ^10^B target (Figure 2a–d), two new bands appeared at 1338.7 and 1200.8 cm^−1^ in the BF_2_ stretching region, which decreased slightly after annealing to 8 K and λ > 300 nm photolysis, and decreased obviously after λ > 220 nm irradiation. In addition, the other new group bands appeared at 1369.4 and 1223.8 cm^−1^ after λ > 220 nm irradiation. In the reaction of ^10^BF_3_ with ^11^B (Figure 2e–h), 1369.4 and 1223.8 cm^−1^ bands appeared on deposition, which decreased largely upon λ > 220 nm irradiation. Meanwhile, new group bands were located at 1338.7 and 1200.8 cm^−1^ after λ > 220 nm photolysis. Obviously, we obtained the same two isomers in ^10^BF_3_ + ^11^B and ^11^BF_3_ + ^10^B experiments.

It is very interesting to observe that in the reaction of ^11^BF_3_ with ^10^B (Figure 2a–d), F_2_^11^B^10^BF was produced at first, but decreased with the emergence of F_2_^10^B^11^BF on >220 nm irradiation. Similarly, in Figure 2e–h, F_2_^10^B^11^BF was observed firstly and then F_2_^11^B^10^BF appeared accompanied with no obvious change of F_2_^10^B^11^BF on the >220 nm irradiation. Apparently, α-F transfer happened between the two species due to the photo irradiation.

This assignment is supported by our DFT frequency calculations (Table 1). The F_2_BBF molecule is predicted to have *C*_s_ symmetry with ^2^A ground state (Figure 4). The anti-symmetric and symmetric B-F stretching modes of the F_2_^11^B(^10^B)-^11^BF molecule were predicted at 1332.3 (1378.8) and 1188.9 (1221.3) cm^−1^ using the B3LYP functional, very close to our observed values of 1336.9 (1369.4) and 1202.4 (1223.8) cm^−1^. Furthermore, the anti-symmetric and symmetric B-F stretching modes of the F_2_^11^B(^10^B)-^10^BF molecule were predicted at 1332.3 (1378.9) and 1192.3 (1226.3) cm^−1^, which are in good agreement with our observed values at 1338.7 (1370.6) and 1200.8 (1241.6) cm^−1^. The results of BPW91 and CCSD(T) are consistent with that of B3LYP.

### 2.2. F_2_BAlF

As shown in Figure 3 and Table 1, the new product absorptions located at 1281.5, 1180.8 and 819.6 cm^−1^ in the reaction of ^11^BF_3_ with laser-ablated Al atoms, increased on annealing to 8 K, but decreased sharply on the λ = 450 nm irradiation and increased again on annealing to 12 K. These bands shifted to 1324.6, 1217.1 and 819.6 cm^−1^ in the reaction of ^10^BF_3_ with Al atoms, which showed a similar response to kinds of photolysis and annealing.

The absorption bands at 1281.5 and 1180.8 cm^−1^ shifted to 1324.6 and 1217.1 cm^−1^, giving 1.0336 and 1.0307 of ^10^B/^11^B isotopic frequency ratio, which match very well with the F-B-F radical vibration mode [16]. The absorption bands at 819.6 cm^−1^ showed no ^10^B/^11^B isotopic frequency ratio shift, indicating that only Al and F are involved in this mode. It is most likely that this absorption arises from terminal Al–F stretching vibrations. All these indicate that this group band is attributable to the F_2_BAlF molecular.

The B3LYP calculations predict the F_2_BAlF molecule to have *C*_S_ symmetry with ^2^A ground state. The calculated BF_2_ anti-symmetric and symmetric mode using B3LYP is 1312.0 and 1172.2 cm^−1^, being overestimated by about 2.3% and underestimated by about 0.7%, respectively. The calculated Al-F stretching vibration is overestimated by about 1.6%, which fits the observed values very well.

## 3. Reaction Product Comparison and Bonding Consideration

Two stable complexes, F_2_BBF and F_2_BAlF, were calculated by the B3LYP functional and parameters are illustrated in Figure 4. The reaction of laser-ablated B atoms with BF_3_ to produce inserted complex F_2_BBF is exothermic by 54.0 kcal/mol at CCSD(T) level. The subsequent α-F transfer reaction to give FBBF_2_ requires an energy barrier of 31.7 kcal/mol. In our experiments of ^11^BF_3_ with ^10^B or ^10^BF_3_ with ^11^B, only one inserted species was observed first and then α-F transfer occurs upon 220 nm photolysis. In the reaction of BF_3_ with Al atom, the F_2_BAlF molecule is produced with an exothermic 15.5 kcal/mol^−1^ reaction at the CCSD(T) level. However, the FB-AlF_2_ produced by a-F transfer from F_2_B-AlF is endothermic by 6.8 kcal/mol^−1^, which could not be observed in our experiments (Figure 5).

The bond angle of B-B-F for the F_2_B-BF molecule is 142°, which is quite different from the 116° of B-Al-F angle for F_2_B-AlF (Figure 4). As shown in Figure 6a, for the F_2_BBF molecule, the B = B bond is composed of one σ bond with an occupation of 1.94 e and one (p-p) π bond with an occupation of 1.00 e in the plane, which leads to a larger B-B-F bond angle. The effective bond order (EBO) of B-B bond is 1.32 calculated by natural bond orbital (NBO) population analysis. The calculated 1.644 Å of B = B bond length is shorter than that of B-B single bond between 1.819 and 1.859 Å, but longer than 1.561 and 1.590 Å of the B = B double bond length in R(H)B = B(H)R (R = :C{N(2,6- Pri_2_C_6_H_3_)CH}_2_) [17], and in OC(H)B = B(H)CO [18], respectively, affirming that B-B bond order in F_2_B-BF is between one and two. Notice that for the F_2_BAlF molecule, the effective bond order (EBO) of B-Al is 0.96 with an occupation of 1.93 e in the σ character (Figure 6b). The Al atom possesses a single electron (mostly from the *s* orbital) which does not participate in bonding. Although B and Al atom are in the same group, their bonding situation is very different. As shown in Appendix A, for the F_2_BBF molecule, both boron atoms have a good hybridization with the *s* and *p* orbital, while little hybridization occurs between the s and p orbital for the B-Al σ bond in the F_2_BAlF molecule.

The energy decomposition analysis (EDA) can be used in quantitative interpretation of chemical bonds’ formation in terms of three major components (Table 2) [19]. For the F_2_BBF molecule, the EDA shows that the total interaction energy of −148.3 kcal/mol between the F_2_B and BF fragments consists of an attractive electrostatic energy of −53.0 kcal/mol, an orbital interaction energy of −193.9 kcal/mol and a large Pauli repulsion of 97.5 kcal/mol. This interaction energy is bigger than that of the H-H single bond (−112.9 kcal/mol), but smaller than the N-N triple bond’s (−232.2 kcal/mol) [19]. Surprisingly, this interaction between B and B is bigger than that of the triple bond between B and heavier transition metal atom that we observed previously [20]. It is possible to breakdown the orbital term ΔE_orb_ into pairwise orbital contributions of the interacting fragments by EDA–Natural Orbitals for the Chemical Valence (NOCV) method [21,22,23]. Figure 6c clearly depicted the natural orbitals for the chemical valence of F_2_BBF. The σ bond between B and B is mainly caused by the outflow of electrons (most 2*s* electron of B) from FB to B of BF_2_ and then the (p-p) π bond is formed by the outflow of one 2*p* electrons of B of BF_2_ to 2*p* vacant orbital of B of BF. The decomposition of the orbital interaction shows that 42.7% (−82.8 kcal/mol) come from the σ bond, while 53.6% (−104.0 kcal/mol) come from the π bonds, respectively.

For the F_2_BAlF molecular, only one σ bond existed between B and Al and both B and Al atom contribute to this σ bond together (Figure 6d). From Figure 6b, we can observe that the occupation of the σ bond is 1.93 e. Moreover, the Al atom possesses a single electron (most from the s orbital) which did not participate in bonding. Although B and Al atoms are in the same group, their bonding situation is very different. In Appendix A, for the F_2_BBF molecule, both two boron atoms have good hybridization with the s and p orbital. While for the B-Al σ bond, the bonding electron is either from the *s* orbital or from the *p* orbital of the Al atom in a different phase and little hybridization happening between the *s* and *p* orbital. Thus, the 3s electrons of aluminum barely participates in bonding with other atoms and no π bond formed in the F_2_BAlF molecule.

Although the FBAlF_2_ molecule was not observed in the experiment by the α-F transfer, a similar bonding composition could be demonstrated to that of FBBF_2_ (Appendix A). The B-Al is also caused by the outflow of electrons (most 2*s* electron of B) from FB to Al, and then the (p-p) π bond is formed by the inflow of one 3*p* electrons of Al to the 2*p* vacant orbital of B. The total interaction energy of −122.0 kcal/mol between BF and AlF_2_ is very strong, and 29.3% (−39.5 kcal/mol) come from the σ bond and 68.3% (−91.9 kcal/mol) from π bond; thus, compounds with σ-donor and π-acceptor bonding modes formed [20].

For further analyzing the bond character, the atoms in the molecule theory (AIM) analysis were performed (Appendix A). The negative value of local energy density H(r) = −0.12656 and −0.02651 for B-B and B-Al was obtained, respectively. The bond critical point between the B and B atom locates in the negative value of the Laplacian value (∇^2^ρ_cp_ = −0.426); however, this value is slightly positive (∇^2^ρ_cp_ = 0.058) and close to zero between the B and Al atom. Appendix A displayed the color-filled maps of the localized orbital locator (LOL) on the F_2_BM (B, Al) plane. It demonstrated that there are high electron localization regions between boron and metal (B and Al), which indicates the covalent bond character. From B to Al, the BF_2_ antisymmetric and symmetric stretch mode can red-shift from 1336.9 and 1202.4 cm^−1^ to 1281.5 and 1180.8 cm^−1^.

## 4. Experimental and Computational Methods

Laser-ablated B and Al atoms react with ^11^BF_3_ and ^10^BF_3_ in excess neon during condensation at 4 K using a closed-cycle helium refrigerator (Sumitomo Heavy Industries Model SRDK-408D2, Japan). A Nd:YAG laser fundamental (1064 nm, 10 Hz repetition rate with 10 ns pulse width) was focused onto the rotating B, or Al target, and typically 20–30 mJ/pulse was used. The laser-ablated enriched ^10^B (Eagle Pitcher, America, 93.8% ^10^B, 6.2% ^11^B), enriched ^11^B (Eagle Pitcher, 97.5% ^11^B, 2.5% ^10^B), and Al (Alfa Aesar, America, 99.999%) atoms were reacted with ^11^BF_3_ and ^10^BF_3_ purchased from Jinglin (Shanghai, China) Chemical Industry Limited Liability Company (Shanghai, China, chemical purity, ≥99.99%) in excess neon spread uniformly onto the CsI window. Infrared spectra were recorded at a resolution of 0.5 cm^−1^ between 4000 and 400 cm^−1^ using a HgCdTe range B detector. Selected samples were irradiated by a mercury lamp (175 W, without globe) with the aid of glass filters to permit the allowed wavelengths to pass.

All structures were optimized at the BPW91/def2-TZVPP and B3LYP/def2-TZVPP [24,25] basis set via the Gaussian 09 program [26] and CAS(9e, 11o)/def2-TZVP [27,28] and CCSD(T)/def2-TZVP(-f) [29,30,31] basis set via the ORCA 4.0.1 program [32,33]. The single point energy calculations were performed with the correlated molecular orbital theory coupled cluster CCSD(T) [29,30,31] theory. Transition states were optimized with the Rational Function Optimization (RFO) method and were verified to link the desired reactant and product through the intrinsic reaction coordinate (IRC) calculations. Atoms in molecules’ (AIM) [34] analysis was performed to elicit detailed information on the bonding characters with the Multiwfn code [35]. The orbital composition and effective bond order and Wiberg bond order were calculated by a natural bond orbital (NBO) population analysis [26,36]. In addition, ab initio calculations based on the high-level multi-configurational wavefunction method were also performed to obtain the accurate electronic structure information of BF_2_-MF compounds by the ORCA 4.0.1 program [32,33]. CASSCF [27] calculations including three active electrons in eight active orbitals [CAS(3e, 8o)], and NEVPT2 [37,38,39] calculations including three active electrons in four active orbitals [CAS(3e, 4o)] were performed with the def2-TZVP [28] basis set for all atoms. The effect of the dynamic correlation was taken into account by NEVPT2 [37,38,39] on top of the wavefunctions at CASSCF level to obtain more accurate energies. The energy decomposition analysis with the natural orbitals of the chemical valence (EDA-NOCV) method [21,22,23] were carried out with the ADF 2017 program [40] package to study the chemical bonding between the B and Al atoms with the B atom.

## 5. Conclusions

The reaction of laser-ablated B and Al atoms with BF_3_ has been studied by the matrix isolation infrared spectrum and theoretical calculations. The structure and properties of the B_2_F_3_ molecule, which can be drawn as F_2_B-BF, have been investigated. The F-B-F stretching mode was located at 1336.9 and 1202.4 cm^−1^. For comparison, the F_2_BAlF molecule was also investigated and the F-B-F stretching mode was at 1281.5 and 1180.8 cm^−1^. The CASSCF analysis, EDA-NOCV calculation, the theory of atoms-in-molecules (AIM) and localized orbital locator (LOL) are applied to investigate the bonding characters of the B-B and B-Al bond in F_2_BBF and F_2_BAlF molecules. The B-B bond in F_2_BBF favors the one and half bond order, in which two boron atoms have a good hybridization between *s* and *p* orbital. Meanhile, due to little hybridization between *s* and *p* orbital, 3s electrons of aluminum barely participate in bonding with other atoms, thus one bond order is formed for the B-Al bond.

## Figures and Tables

**Figure 1 molecules-28-00554-f001:**
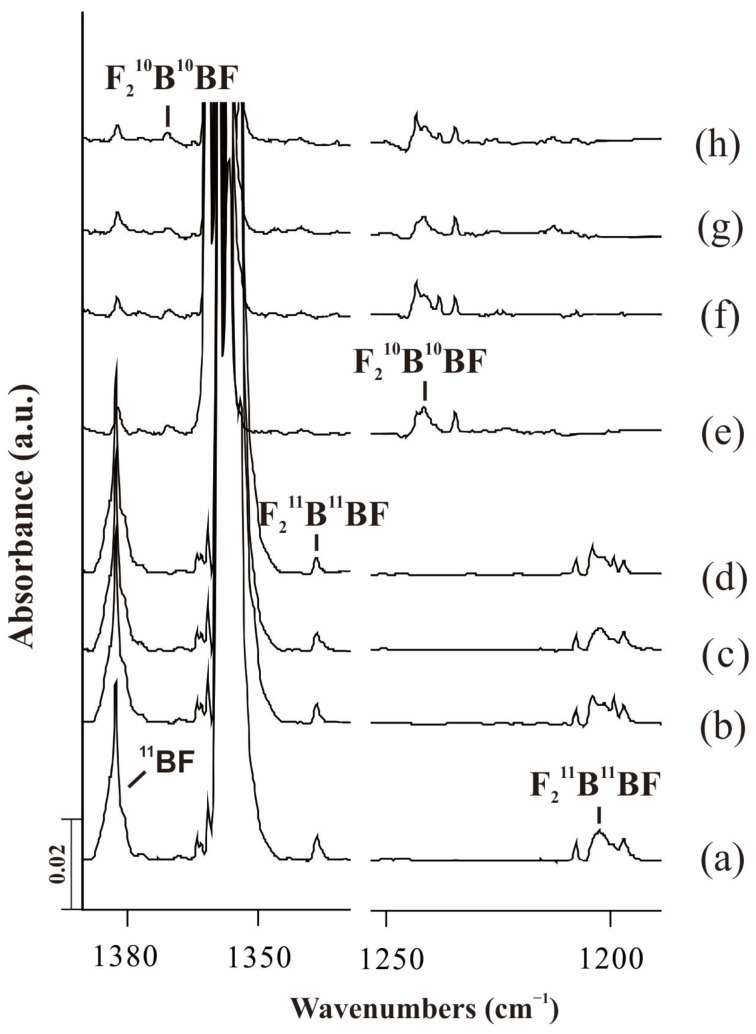
Infrared spectra of the laser-ablated B atoms’ reactions with BF_3_ in excess solid neon. (**a**) co-deposition of ^11^B + 1.0% ^11^BF_3_ for 60 min; (**b**) after annealing to 8 K; (**c**) after λ > 300 nm irradiation for 6 min; (**d**) after annealing to 12 K; (**e**) co-deposition of ^10^B + 0.5% ^10^BF_3_ for 60 min; (**f**) after annealing to 8 K; (**g**) after λ > 300 nm irradiation for 6 min; (**h**) after annealing to 12 K.

**Figure 2 molecules-28-00554-f002:**
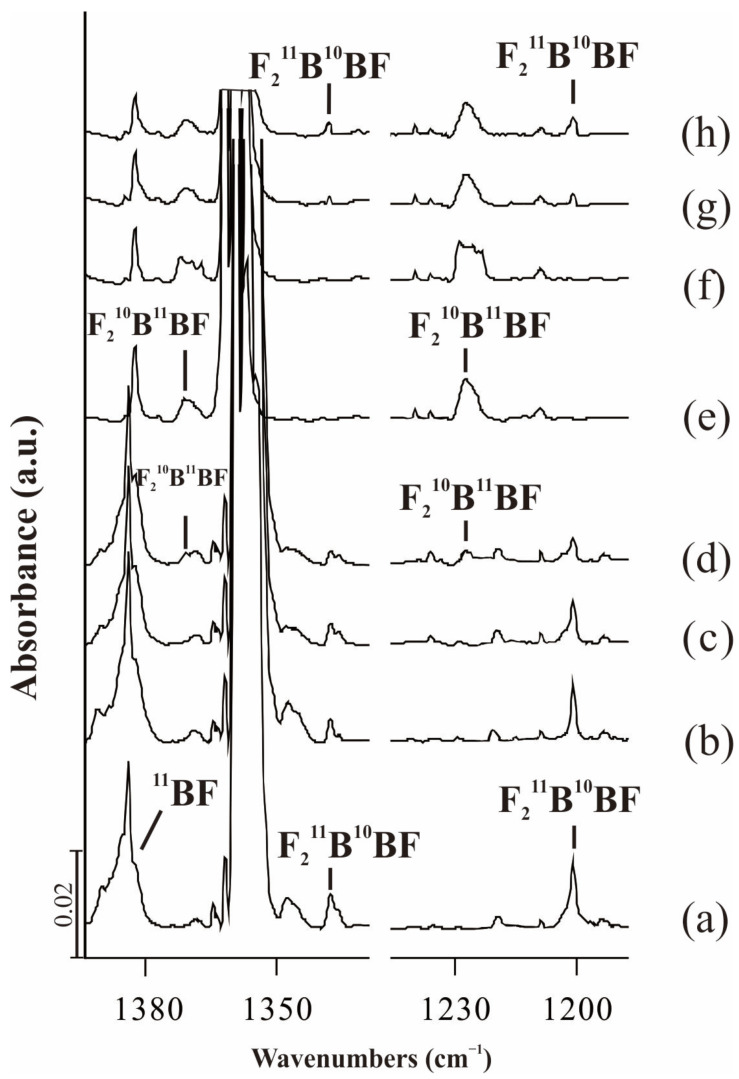
Infrared spectra of the laser-ablated B react with BF_3_ in excess solid neon. (**a**) co-deposition of ^10^B + 1.0% ^11^BF_3_ for 60 min; (**b**) after annealing to 8 K; (**c**) after λ > 300 nm irradiation for 6 min; (**d**) λ > 220 nm irradiation for 6 min; (**e**) co-deposition of ^11^B + 0.5% ^10^BF_3_ for 60 min; (**f**) after annealing to 8 K; (**g**) after λ > 300 nm irradiation for 6 min; (**h**) λ > 220 nm for irradiation for 6 min.

**Figure 3 molecules-28-00554-f003:**
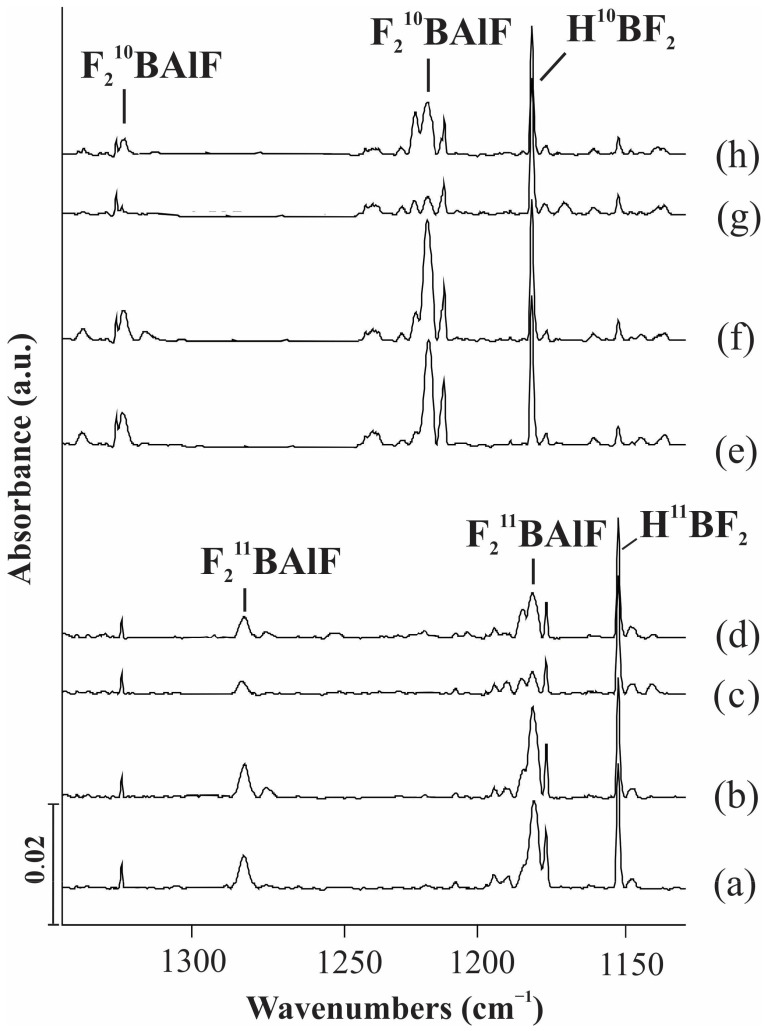
Infrared spectra of the laser-ablated Al atoms reactions with BF_3_ in excess solid neon. (**a**) co-deposition of Al + 0.5% ^11^BF_3_ for 60 min; (**b**) after annealing to 8 K; (**c**) after λ = 450 nm irradiation for 6 min; (**d**) after annealing to 12 K; (**e**) co-deposition of Al + 0.5% ^10^BF_3_ for 60 min; (**f**) after annealing to 8 K; (**g**) after λ = 450 nm irradiation for 6 min; (**h**) after annealing to 12 K.

**Figure 4 molecules-28-00554-f004:**
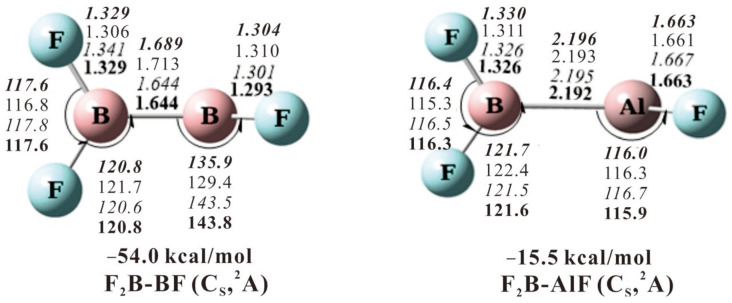
Structures of the product F_2_BMF (M = B, Al) optimized using the CCSD(T)/def2-TZVP(-f) (**bold** and *italic*), CAS(9e, 11o)/def2-TZVP, BPW91/def2-TZVPP (*italic*) and B3LYP/def2-TZVPP (**bold**) functionals/basis set. The def2-TZVP(-f), def2-TZVP and def2-TZVPP basis are set for all atoms. Bond lengths are in Å and angles in degrees. The energies are in kcal/mol and relative to corresponding M + BF_3_ calculated by CCSD(T)/def2-TZVPP.

**Figure 5 molecules-28-00554-f005:**
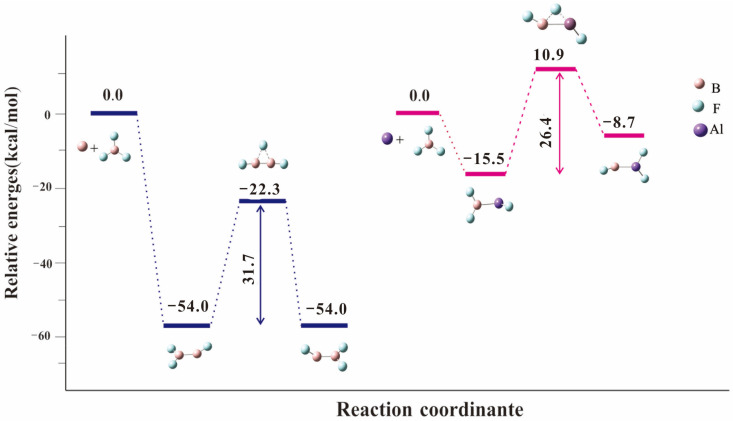
Potential energy surface of group 13 atoms (B and Al) and BF_3_ reaction products calculated at CCSD (T)/def2-TZVPP level. Energies were given in kcal/mol.

**Figure 6 molecules-28-00554-f006:**
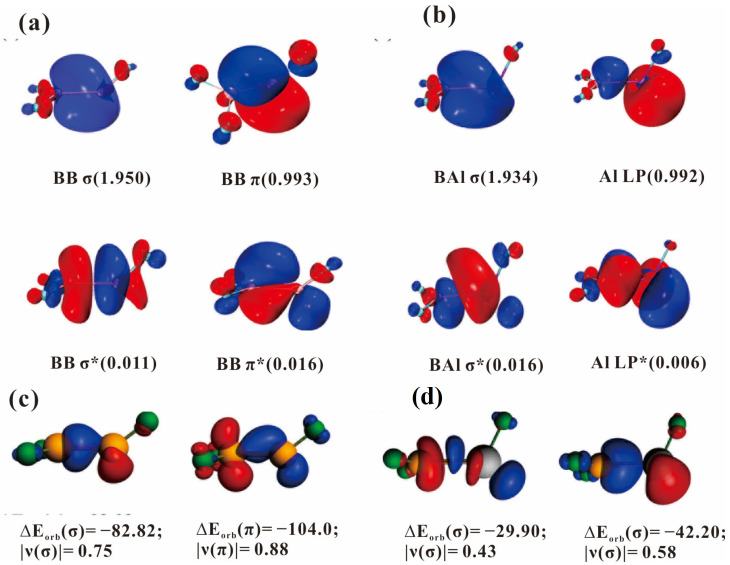
(**a**) The active molecular orbitals of F_2_BBF at CASSCF (3e, 8o)/def2-TZVP level. The isosurface value is 0.04 a.u. MO occupation numbers are given with each orbital; (**b**) The active molecular orbitals of F_2_BAlF at CASSCF (3e, 8o)/def2-TZVP level. The isosurface value is 0.04 a.u. MO occupation numbers are given with each orbital; (**c**) Plot of the deformation densities Δρ of the F_2_B→BF σ donation and BF→BF_2_ π back-donation in F_2_BBF with the associated interaction energy ∆Eorb and charge eigenvalues |ν_n_|(in e). The charge flow is from red → blue; (**d**) Plot of the deformation densities Δρ of the F_2_B→AlF σ donation and AlF→BF_2_ σ back-donation in F_2_BAlF with the associated interaction energy ∆Eorb and charge eigenvalues |ν_n_|(in e). The charge flow is from red → blue.

**Table 1 molecules-28-00554-t001:** Observed and calculated fundamental frequencies of F_2_BBF and F_2_AlBF isotopomers in the ground ^2^A state.

Approximate Description	Obs(Ne)	Cal(int) ^a^	Cal(int) ^b^	Cal(int) ^c^	Obs(Ne)	Cal(int) ^a^	Cal(int) ^b^	Cal(int) ^c^
BBFantisymstr	/	1491.2(52)	1461.7(50)	1466.2	/	1501.4(41)	1471.7(37)	1477.8
BF_2_antisymstr	1336.9	1332.3(294)	1286.7(265)	1376.3	1369.4	1378.8(317)	1331.6(284)	1425.1
BF_2_symstr	1202.4	1188.9(465)	1150.9(427)	1213.6	1223.8	1221.3(505)	1182.6(465)	1245.4
		F_2_^11^B^10^BF				F_2_^10^B^10^BF		
BBFantisymstr	/	1538.6(70)	1508.0(63)	1511.4	/	1546.82(56)	1516.6(49)	1520.8
BF_2_antisymstr	1338.7	1332.3(294)	1286.7(265)	1376.3	1370.6	1378.90(324)	1331.6(284)	1425.1
BF_2_symstr	1200.8	1192.3(462)	1154.2(424)	1217.9	1241.6	1226.32(505)	1187.3(463)	1251.9
		F_2_^11^BAlF				F_2_^10^BAlF		
BF_2_antisymmstr	1281.5	1312.0(277)	1270.9(257)	1342.1	1324.6	1357.4(299)	1314.9(275)	1389.1
BF_2_symstr	1180.8	1172.2(320)	1133.9(309)	1190.3	1217.1	1208.2(378)	1168.8(336)	1227.4
AlFstr	819.6	806.7(106)	780.1(98)	814.5	819.6	806.8(106)	780.2(98)	814.6

The vibrational frequencies (cm^−1^) and intensities (km/mol, in parentheses) are calculated using: ^a^ B3LYP/def2-TZVPP; ^b^ BPW91/def2-TZVPP; ^c^ CCSD(T)/def2-TZVP(-f) functionals/basis set.

**Table 2 molecules-28-00554-t002:** The results of EDA-NOCV theory for F_2_BMF at B3LYP/TZ2P.

Orbitals	F_2_BBF (^2^A)	Orbitals	F_2_BAlF (^2^A)
Δ*E*_int_	−148.3	Δ*E*_int_	−28.84
Δ*E*_Pauli_	97.5	Δ*E*_Pauli_	111.5
Δ*E*_elstat_	−53.0 (21.5%)	Δ*E*_elstat_	−60.4 (42.8%)
Δ*E*_orb_	−193.9 (78.5%)	Δ*E*_orb_	−80.7 (57.2%)
Δ*E*_σ_	−82.82 (42.7%)	Δ*E*_σ_	−29.9 (37.1%)
Δ*E*_π_	−104.0 (53.6%)	Δ*E*_σ_	−42.2 (52.3%)
Δ*E*_orb_(rest)	−7.1 (3.7%)	Δ*E*_orb_(rest)	−8.6 (15.6%)
Δ*E*_dist_	1.1	Δ*E*_dist_	0.76

Energy values are given in kcal/mol.

## Data Availability

If you interested in our article and could not find some other data, please contact us from email.

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
