# Peer review of "F2BMF (M = B and Al) Molecules: A Matrix Infrared Spectra and Theoretical Calculations Investigation"

_molecules, 2023, doi:10.3390/molecules28020554_

Round 1

Reviewer 1 Report

It is an excellent paper; my comments and questions will focus on the theoretical part since it is my area of expertise. 

1) I strongly suggest changing the functional (not function as mentioned in the main text) to something else since B3LYP was parameterized for organic systems. There are many other options on the market. The authors can do the CCSD(T) calculations on top of these new geometries.

2) I am surprised that if the authors carry out an EDA, why don't they optimize ADF molecules where there are other choices of functionals? 

3) I also don't understand why the authors carry out CASSCF calculations. Why they do not discuss them or take these as the basis for these calculations to discuss geometry and energetics? Do the systems have a multireference character or not? Employing this type of methodology just to calculate EBO is not the best practice. 

4) From the point of view of electron density analysis, how do the authors claim that "for B-B and B-Al respectively indicates the covalent bond character apparently"? It is impossible to conclude from the value of the Laplacian or H. 

In summary, the authors must choose the level they will use to discuss the results correctly. If other methods are used, it is necessary to have a reason to do so. If the systems present a multireference character, then CASSCF is justified. For these reasons, the article requires further revision. 

Author Response

Reviewer 1:It is an excellent paper; my comments and questions will focus on the theoretical part since it is my area of expertise. 

Q1. I strongly suggest changing the functional (not function as mentioned in the main text) to something else since B3LYP was parameterized for organic systems. There are many other options on the market. The authors can do the CCSD(T) calculations on top of these new geometries.

Reply: B3LYP is currently very popular functional, generally. If the optimized system of main group do not significantly involve weak interactions and contain large conjugate features, reasonable results calculated at B3LYP level can be obtained. There are substantial references to provide support. In addition, BPW91 functional was also carried out, the geometry and frequencies were shown in Figure 4, Table 1 and Table S3 which also matched the experimental data very well. CCSD(T) calculations are also carried out for our systems, the information of geometry and main frequencies are shown in Figure 4 and Table 1.

Q2. I am surprised that if the authors carry out an EDA, why don't they optimize ADF molecules where there are other choices of functionals? 

Reply: Due to many Gaussian basis set are not supported by ADF, such as 6-31G*, def2-TZVP, cc-pVTZ, so we optimize geometries by Gaussian 09 program, and then EDA analysis were carried out with the ADF program.

Q3. I also don't understand why the authors carry out CASSCF calculations. Why they do not discuss them or take these as the basis for these calculations to discuss geometry and energetics? Do the systems have a multireference character or not? Employing this type of methodology just to calculate EBO is not the best practice. 

Reply: We carry out CASSCF calculations in order to provide more information. When the HOMO-LUMO gap is small, the systems will have strong multireference character. For F2BBF (2A), the HOMO-LUMO gap of α-orbital is 0.14 ev and that of β-orbital is 0.30 ev. For F2BAlF (2A) the values are 0.11 ev and 0.28 ev respectively. The small gap indicates the systems have a multireference character. In general DFT have some defect to deal with multireference system so we use CASSCF calculation to patch up it. But we found that the results obtained by the two methods are consistent. The original EBO is not the best practice, we recalculated effective bond order (EBO) by a natural bond orbital (NBO) population analysis which shown in in Table S3.

Q4. From the point of view of electron density analysis, how do the authors claim that "for B-B and B-Al respectively indicates the covalent bond character apparently"? It is impossible to conclude from the value of the Laplacian or H. 

Reply: For further expounding our conclusion that "for B-B and B-Al respectively indicates the covalent bond character apparently", we employed localized orbital locator (LOL) which the supplement information can be seen from line 204 to line 208 and Figure S3. Although negative value of local energy density H(r) = -0.12656 and -0.02651 for B-B and B-Al respectively, Figure S3 displayed the color-filled maps of localized orbital locator (LOL) on the F2BM (B, Al) plane. It suggested that there were high electron localization region on the B-M bond, which indicates the covalent bond character.

Q5. In summary, the authors must choose the level they will use to discuss the results correctly. If other methods are used, it is necessary to have a reason to do so. If the systems present a multireference character, then CASSCF is justified. For these reasons, the article requires further revision. 

Reply: Yes. We have revised it. In general DFT have some defect to deal with multireference system so we use CASSCF calculation to patch up it. But we found that the results obtained by the two methods are consistent. So we keep two results of these two methods.

Reviewer 2 Report

Present work is devoted to cooperative theoretical and experimental investigation of F2BMF (M=B and Al) molecules. In general, it is fruitful work with complicated experimental part and varied theoretical part.

While reading the paper, the several questions/comments were raised.

Major Comments:

1. Conclusion part should be improved. It is not enough only to highlight that several investigations devoted to B-B and B-Al bonding peculiarities were done. It is important to reveal what is the result of given investigations.

2. I don't see the supplementary part of the work. If it is not a failure of susy platform, auhor should upload the supplementary part.

3. Where is information about Wiberg bond order analysis? In Experimental and computational methods part author noted that the orbital composition and Wiberg bond index analyses were calculated by a natural bond orbital (NBO) population analysis. But in main text of work this result was not mentioned.

4. Table 1 should be improved. If it is possible, the core information should be presented in graphical form (diagram, plot etc). Residual information should be moved to the supplementary part of the work. 

5.  What is the yield of target products? Are there techniques to estimate the yield of the target? What about the stability of present molecules?

Minor comments:

1. 0.42kcalmol-1 should be changed to 0.42kcalmol-1.

2. ab initio should be marked italicized (ab initio).

3. B3LYP function should be changed to B3LYP functional.

4. For the statement "B=B double bond length 1.561 Å in R(H)B=B(H)R (R = :C{N(2,6- Pri2C6H3)CH}2)" appropriate reference should be added.

5. What software was used for performing EDA–Natural Orbitals for Chemical Valence? Appropriate information should be added to Experimental and computational methods part.

Overall, present work can be suitable for publication after careful major revision.

Author Response

Reviewer 2: Present work is devoted to cooperative theoretical and experimental investigation of F2BMF (M=B and Al) molecules. In general, it is fruitful work with complicated experimental part and varied theoretical part. While reading the paper, the several questions/comments were raised.

Q1. Conclusion part should be improved. It is not enough only to highlight that several investigations devoted to B-B and B-Al bonding peculiarities were done. It is important to reveal what is the result of given investigations.

Reply: Revised. We have added the detail of investigation in this part. (Line 243-252)

Q2. I don't see the supplementary part of the work. If it is not a failure of susy platform, auhor should upload the supplementary part.

Reply: Supplementary part of the work have been initially provided, and we will provide it again later.

Q3. Where is information about Wiberg bond order analysis? In Experimental and computational methods part author noted that the orbital composition and Wiberg bond index analyses were calculated by a natural bond orbital (NBO) population analysis. But in main text of work this result was not mentioned.

Reply: We add the information about bond order analysis in Table S2. Because the values of Wiberg bond order are too small, we discuss bond order with EBO.

Q4. Table 1 should be improved. If it is possible, the core information should be presented in graphical form (diagram, plot etc). Residual information should be moved to the supplementary part of the work. 

Reply: Thank you for your advice. We improved the Table 1 concisely, and the residual information have been moved to the supplementary part of the work. 

Q5.  What is the yield of target products? Are there techniques to estimate the yield of the target? What about the stability of present molecules?

Reply: Unfortunately due to the limitation of the experimental conditions, we are unable to know the exact yield. The binding energy can be employed to measure the stability of products. The binding energy of F2BBF (2A) with respect to ground-state F2B and BF molecules was calculated to be -47.1 kcal/mol with CCSD(T), and the binding energy of F2BAlF (2A) with respect to ground-state F2B and AlF molecules was -21.9 kcal/mol. The negative binding energy indicates the stability of F2BBF and F2BAlF molecules.

Q6. Minor comments:

  1. 0.42kcalmol-1 should be changed to 0.42kcal⋅mol-1.

Reply: Revised. (Line 31)

2). ab initio should be marked italicized (ab initio).

Reply: Revised. (Line 48, 232, 289, 339)

  1. B3LYP function should be changed to B3LYP functional.

Reply: Revised. (Line 97, Line 134)

  1. For the statement "B=B double bond length 1.561 Å in R(H)B=B(H)R (R = :C{N(2,6- Pri2C6H3)CH}2)" appropriate reference should be added.

Reply: Revised. (Reference 18)

  1. What software was used for performing EDA–Natural Orbitals for Chemical Valence? Appropriate information should be added to Experimental and computational methods part. Overall, present work can be suitable for publication after careful major revision.

Reply: Revised. (Line 239 to line 241)

Round 2

Reviewer 1 Report

The fact that a method is popular does not mean that it is correct for describing specific systems. As a theoretician, I cannot accept that this is the justification for using B3LYP and much less when the authors mention that the systems have a particular multi-referential character. Therefore, this article cannot be accepted as it is. The experimental work is excellent, and the systems are exciting, so why risk it? Change the method to an adequate one, not the one that is popular and that everybody uses even though it is incorrect. I leave the decision to the editor.

Author Response

Reply: We are very sorry for our reply because it cause some mislead to reviewer. We have added CASSCF calculation to our manuscript including geometry and frequencies optimization (Figure 4 and Table S3) and corresponding discussion have also been revised.

For geometry optimization, there is no big difference between CASSCF and other methods including B3LYP, BPW91 and CCSD (T). But for the frequencies optimization, there is huge error between observed value and CASSCF value. Specifically, for F211B(10B)-11BF molecule, the value predicted by CASSCF method is 1425.9 (1453.7) and 1261.1(1287.1) cm-1 for anti-symmetric and symmetric B-F stretching modes, which is overestimated by about 93.6 cm-1 (74.9 cm-1) and 72.2 cm-1 (65.8 cm-1) respectively. For F211B(10B)-10BF molecule, the predicted value calculated by CASSCF is also overestimated by 98.0 cm-1 (107.6 cm-1) and 69.7 cm-1 (59.5 cm-1). Similarly, for the F2BAlF molecule, the calculated BF2 anti-symmetric and symmetric mode is at 1411.1 and 1277.2 cm-1 by CASSCF method, being overestimated by about 129.5 cm-1 and 96.4 cm-1respectively. Basically, the error is more than 5%, and some of them reach 9%. So we put CASSCF value in Table S3. Compared with CASSCF results, the predicted frequencies from B3LYP is not more than 3%.

In general, we use B3LYP and BPW91 method to optimize the structures and calculate the frequencies, which often give very close frequency value to our observed value in experiment. Then we use CASSCF functional to analyze the active orbitals and use CCSD(T) method to calculate the relative energy. In our previous reports, we have adopted similar strategy to analyze some systems. For example, Inorg. Chem. 2019, 58, 2363-2371; J. Phys. Chem. A. 2018, 122, 7301-7311; Inorg. Chem. 2019, 58, 13418−13425.

Reviewer 2 Report

After all corrections manuscript is suitable for publication.

Author Response

Reviewer 2:After all corrections manuscript is suitable for publication.

Reply: Revised.